# Core Promoter Regions of Antisense and Long Intergenic Non-Coding RNAs

**DOI:** 10.3390/ijms24098199

**Published:** 2023-05-03

**Authors:** Ekaterina A. Savina, Tatiana G. Shumilina, Vladimir G. Tumanyan, Anastasia A. Anashkina, Irina A. Il’icheva

**Affiliations:** 1Engelhardt Institute of Molecular Biology, Russian Academy of Sciences, Vavilova, 32, Moscow 119991, Russia; 2Department of Information and Internet Technologies of the Institute of Digital Medicine, Sechenov University, 8-2 Trubetskaya str., Moscow 119991, Russia

**Keywords:** eukaryotes, long non-coding RNAs, long intergenic non-coding RNAs, antisense RNAs, RNAP II transcription apparatus, RNAP II core promoter, DNA local structure, ultrasonic cleavage, DNase I cleavage, TATA box

## Abstract

RNA polymerase II (POL II) is responsible for the transcription of messenger RNAs (mRNAs) and long non-coding RNAs (lncRNAs). Previously, we have shown the evolutionary invariance of the structural features of DNA in the POL II core promoters of the precursors of mRNAs. In this work, we have analyzed the POL II core promoters of the precursors of lncRNAs in *Homo sapiens* and *Mus musculus* genomes. Structural analysis of nucleotide sequences in positions −50, +30 bp in relation to the TSS have shown the extremely heterogeneous 3D structure that includes two singular regions - hexanucleotide “INR” around the TSS and octanucleotide “TATA-box” at around ~−28 bp upstream. Thus, the 3D structure of core promoters of lncRNA resembles the architecture of the core promoters of mRNAs; however, textual analysis revealed differences between promoters of lncRNAs and promoters of mRNAs, which lies in their textual characteristics; namely, the informational entropy at each position of the nucleotide text of lncRNA core promoters (by the exception of singular regions) is significantly higher than that of the mRNA core promoters. Another distinguishing feature of lncRNA is the extremely rare occurrence in the TATA box of octanucleotides with the consensus sequence. These textual differences can significantly affect the efficiency of the transcription of lncRNAs.

## 1. Introduction

An analysis of the genome-wide catalog of human transcripts obtained within the ENCODE project [1] led to the conclusion that the transcribed part of the human genome is at least 75% of its length. The set of transcripts contains mRNA precursors, whose exons encode proteins after processing, make up less than 3% of transcripts and RNA molecules that do not encode proteins and 97% of all transcripts [2,3,4]. The transcription of the eukaryotic nuclear genome, which results in the formation of various RNA variants, is carried out by three different RNA polymerases: RNA polymerase I (POL I), which transcribes the 45S precursor of ribosomal RNA (rRNA)—5.8S, 18S, and 28S; POL II, which transcribes precursors for mRNAs and for a number of non-coding RNAs (ncRNAs); and RNA polymerase III (POL III), which transcribes precursors for tRNA, 5S rRNA, 7SL RNA, and some small RNAs.

NcRNAs vary considerably in length, shape, and functionality. The lncRNAs POL II transcribes at various elements of the eukaryotic genome, including promoters, enhancers, and intergenic regions. They can be considered as precursors of key regulators of the gene expression network [5]. Additionally, it is quite possible that far from all the functions of the ncRNA molecules present in the cell have been characterized as of now.

The editing of primary RNAs transcripts, including ncRNAs, plays relevant roles in the molecular processes in living cells. Disturbances in editing may be the cause of pathogenic processes. For example, “change of the binding capacity of 359 miRNAs towards 9654 target genes” as the result of oxidative stress is found to be coupled with retinal degeneration [6].

Since long intergenic non-coding RNAs (lincRNAs) and antisense RNAs (asRNAs), like mRNAs, are transcribed by the POL II apparatus, many features of their biogenesis should be similar; however, these two classes of transcripts differ in many respects. Compared to mRNAs, lncRNAs are known to have a low interspecies conservatism; they have a low expression level and were previously considered as transcriptional noise. Their functional purpose has been questioned for a long time [7]. They also differ from mRNAs in their markedly greater tissue specificity [8].

Structural features of DNA that determine POL II core promoters of lncRNAs, namely lincRNAs and asRNAs, have not been studied yet. In this work, we have done statistical analysis of the structural and textual characteristics of nucleotide sequences of core promoters for asRNAs and lincRNAs for *H. sapiens* and *M. musculus*.

## 2. Results

We have used the first release of collections from the new section of the Eukaryotic Promoter Database (EPD) (http://epd.vital-it.ch (accessed on 24 October 2022)) [9]. This resource allows access to a collection of databases of experimentally validated promoters of several model organisms, for which TSS mapping was the result of high-throughput experiments such as CAGE and Oligo-capping, resulting in high precision and high coverage. It is based on the HGNC, Ensembl and RefSeq gene annotation and uses the TSS mapping data from both the ENCODE and FANTOM5 consortia.

The database of *H. sapiens* POL II non-coding promoters contain 2339 samples, and the database of *M. musculus* POL II non-coding promoters contain 3077 samples. These databases include the promoters of lincRNAs and asRNAs. The profiles of averaged textual, structural, mechanical, and physicochemical properties of 80 bp core promoter sequences (in positions from -50 to +30) in relation to the TSS were constructed. We checked that all these sequences are 80 nucleotides long and strictly defined.

### 2.1. Textual Characteristics of the Nucleotide Sequences Forming Core Promoters of Non-Coding RNAs in H. sapiens and M. musculus

First, we compared the percentages of A, T, G, and C nucleotides in the non-coding core promoters of *H. sapiens* and *M. musculus*. For simplicity, in accordance with IUPAC nomenclature, we will use the terms W (for nucleotides A and T) and S (for nucleotides G and C). The frequencies of nucleotide occurrence at each position along the coding strands of *H. sapiens* are shown in Figure 1a and those along the coding strands of *M. musculus* in Figure 1b. For both organisms, the percentage of S exceeds that of W in all positions with the exception of the TATA element, where the percentages of W are almost equal to that of S in the promoters of *H. sapiens*, while in the promoters of *M. musculus*, T even exceeds A, G, and C. These are in line with the data on the mammalian promoters of mRNA [10,11].

The logo representation of non-coding promoter sequences in comparison with mRNA promoter sequences with information content of 0.4 bits is shown in Figure 2a,b for *H. sapiens* and in Figure 3a,b for *M. musculus*. Logos were made at http://weblogo.threeplusone.com (accessed on 24 October 2022). The remarkably lower information content of the majority of the positions of non-coding promoters is a distinctive feature of both logo profiles presenting non-coding promoter sequences. The exception is the hexanucleotide of the INR element, where informational content of non-coding promoters exceed those of mRNA promoters and goes beyond 0.4. Therefore, logo representation with information content of 0.6 bits for both non-coding promoters is shown in Appendix A. All other positions of the non-coding promoters either do not carry any textual information, or its informational content does not exceed 0.05 bits. So, various motifs in non-coding promoters are very weakly expressed. It can be assumed that they do not play a significant role in the regulation of the transcription of non-coding RNAs.

Table 1 presents octanucleotides with leading frequencies in the position of the TATA element of non-coding promoters of *H. sapiens* and *M. musculus*. One can see that the frequencies of occurrence of these octanucleotides are rather close. This result is in line with what we obtained from the analysis of mRNA promoters [10,11]. The leading frequencies of occurrence of octanucleotides do not exceed 0.5%. The leading octanucleotides in *H. sapiens* TATA element are GTTTTCA (0.47%) and AGAATAAA (0.38%), and in the *M. musculus* TATA element, TTTTTTTT (0.42%). There are few octanucleotides that do correspond to the TATAWAAR consensus. Namely, in non-coding promoters of *H. sapiens*, this consensus is presented by TATAAAAG (0.17%) and TATATAAA (0.13%), and for *M. musculus*, TATATAAG (0.10%). As it is seen from Table 1, in the non-coding promoters in the TATA box, there are few hexanucleotides with the TATAWAAR consensus. This position is occupied by those octanucleotides that are capable of bending towards the major groove with low energy costs. They may not meet the TATAWAAR consensus.

We limited the samples by the condition “*restricting the selection to promoters that contain a TATA box*” and obtained a set of 250 promoters for *H. sapiens* (this is 10.7% of the 124 total *H. sapiens* sample) and a set of 311 promoters for *M. musculus* (this is 10.1% of the total *M. musculus* sample). The first 20 octanucleotides with maximum frequencies in their sets are shown in Table 2. The leading octanucleotides in the *H. sapiens* set are AGAATAAA (3.6%) and TTTTATAA (2.4%), and in the *M. musculus* set, ACAATATA (1.61%). In our opinion, most of the octanucleotides that are present in Table 2 only faintly resemble the “TATAWAAR” consensus. In the set of *M. musculus*, there are only three consensus sequences (TATAAAAG, 0.96%; TATATAAG, 0.96%; and TATAAAAA, 0.64%). In the set of *H. sapiens* they are TATAAAAG, 1.60% and TATATAAA 1.20%.

For the analyzed position for the TATA box, we have chosen in the region −31–−24 bp in relation to the start of transcription. It was chosen because, namely in this region, we meet the minimum in the profiles of the physical parameter “Stacking energy” and the maximum in the profiles of the physical parameter “Mobility to bend towards major groove” (Figure 4a,f). Since, at this position, we found only few octanucleotides that satisfy the TATA consensus, we decided to analyze positions shifted one or two steps to the right and to the left. The results are presented in Appendix A. Such an analysis did not give fundamentally different results.

Crystallographic studies have shown that TBP–DNA complex formation is permitted on many A/T-rich promoter sequences, albeit with dramatically reduced transcriptional efficiency in some cases and diffraction-quality co-crystals having been obtained with C:G or G:C base pairs at four of the five TATA box positions [12]. Therefore, it is not surprising that we encounter a wide variety of octanucleotides in the position of the TATA box.

We have shown earlier [11] that the *H. sapiens* and *M. musculus* core promoters of mRNA also contain various octanucleotides in the TATA position, while octanucleotides that do correspond to the TATAWAAR consensus are preferred; In non-coding promoters, it is not so. We speculate that this may influence the lower transcriptional efficiency of non-coding promoters than that of the mRNA promoters.

The INR element of non-coding RNA promoters of both organisms, unlike other promoter positions, is highly selective for nucleotide sequence (Figure 2 and Figure 3). Their content in terms of dinucleotides PyPu, PuPu, PyPy, and PuPy in positions −1, +1 is shown in Table 3, while the frequencies of occurrence of each of 16 dinucleotides is shown in Table 4, and the frequencies of occurrence of tetranucleotides in positions −2, +2 is shown in Appendix A.

One can see a preference for PyPu in positions −1, +1. The most rarely observed in these positions are the dinucleotides PuPy (Table 3). In the two organisms, the dinucleotide CA and the tetranucleotide TCAG in the coding strand in this position are preferable (Table 4 and Appendix A). The same preferences was found in mRNA promoters earlier [10,11]. Some minor differences in the percentage of dinucleotides of the PyPu group are observed between the mRNAs and the non-coding RNAs promoters. Namely, the percentage of TG is higher in the non-coding RNA promoters of *H. sapiens* than in the mRNA promoters. We have speculated that T/G-rich promoters present as promoters of asRNAs and are copied from the template chain of the DNA.

What properties of PyPu dinucleotides and especially the CA dinucleotide determine their preference in the position −1, +1? We have discussed this question in detail earlier [11]; however, let us briefly reiterate these considerations. This position is in the center of the INR element, which is responsible for the double helix diverge. It is known that deformability of the dinucleotides decreases in the row PyPu > PuPu > PuPy. This was shown with the help of spin probe while studying the effects of nucleotide sequence on DNA duplex dynamics [13]. The special mobility of the PyPu steps is explained by the greater intensity of S↔N dynamics in the furanose cycles in 5′-terminal pyrimidines, compared to those in 5′-terminal purines, and after 5′Cyt, it reaches its maximum [14]. The advantage of the CpA step over the CpG step in the positions −1, +1 can be explained by the presence of only two hydrogen bonds, which must be broken at the initial stage of chain divergence. The reactivity of d(CA/TG) repeats to the conformation-sensitive reagent chloroacetaldehyde, which reacts with unpaired adenines and cytosines, experimentally explains the unique features of the CA/TG step [15].

### 2.2. Physical and Structural Anisotropy of the Naked DNA in the Non-Coding Core Promoters

Differences in the physical and structural properties of the individual base steps, namely double-stranded dinucleotides, lead to local variations on the profiles of physical and structural properties of the fragments of naked double-stranded DNA. For example, bending anisotropy is sequence-dependent and, to a first approximation, reflects both the geometry and stability of the individual base steps [16].

We have built profiles of the base step characteristics for the sets of the non-coding core promoters of *H. sapiens* and *M. musculus* using indices of numerical parameterization of structural and physical characteristics for the ten double-stranded dinucleotides. They are collected in the DiProDB database (http://diprodb.fli-leibniz.de (accessed on 24 October 2022)) [17]. There are parameters of many different properties of double-stranded dinucleotides in this database. Six of them are most suitable for assessing the effect of the nucleotide sequence on the anisotropy of the fragment with respect to the bending of the helix axis. Namely, we have used Stacking energy, which is a part of an enthalpy of DNA formation and defines its stabilizing forces, and the Roll, which defines an angle between the average planes of two neighboring base pairs, and its positive value corresponds to the opening of this angle towards the minor groove. Therefore, among the three rotational parameters (helical Twist, Roll and Tilt) the Roll is the most important for understanding the bending of DNA [18]. The Slide defines the mutual displacement of the neighboring base pairs in the direction perpendicular to the minor and major grooves. Positive Slide values are a distinguishing feature of B-DNA, while in the A-form of DNA, the values of the Slide are always negative. Thus, the sign of the Slide is an important indicator that allows for discriminating B- and A-DNA forms [19,20]. The remaining three parameters are the rigidity of the double helix in relation to changes from various equilibrium structural parameters. They are the Stiffness of the structure to the Roll alteration, the Stiffness of the structure to the Slide alteration, and the Mobility to bend towards the major groove.

The database contains several versions of parameterization of the same name properties, and earlier [10], we verified that the profiles built from different versions of the parameters are in qualitative agreement with each other. The profiles of these physical and structural parameters are presented in Figure 4a–f. We present the profiles of the variations of the Stacking energy (Figure 4a) and the base-pair step parameters of Roll and Slide (Figure 4b,d) in the parameterization of Perez et al. [21], and the profiles of the Stiffness variation in the DNA double helix to the Roll and Slide alterations (Figure 4c,e) in the parameterization of Goni et al. [22]. These five parameters describe DNA at the base-pair step resolution. To evaluate the Stiffness of the structure to bend towards the major groove, we used the parameterization of Gartenberg and Crothers [23]. Their parameter, “Mobility to bend towards major groove”, was resolved for all 16 dinucleotides and related to each of the complementary strands. In Figure 4f this characteristic is presented in the upper strand (the strand complementary to the template, namely the “coding strand”). While Figure 4a–f presents the profiles of the characteristics of the core region (−50–−30) of non-coding promoters for both organisms, Appendix A presents the profiles of the same characteristics in the *H. sapiens* genomic sequences from the regions (−499–−420) and (−299–−220) in relation to the start of transcription as well as the profiles of the 80 bp set of 3000 computer-simulated random nucleotide sequences. They are presented along with the eponymous profiles of the *H. sapiens* non-coding core promoter sequences.

The stacking energy in core regions of the promoter sequences of non-coding RNAs of *H. sapiens* and *M. musculus* is about −16.5 ± 0.1 kcal/mol (Figure 4a), which is close to the value characteristic of mRNA core promoters of mammals [10,11]. There are shallow global minimums on the profiles of both organisms between the positions −24 bp and −31 bp relative to the TSS. The Roll and Slide values of the naked DNA belongs to the B-family with the exception of the INR element. The values of the Roll are about 1.5°, and a shallow global minimum can also be seen between the positions −24 bp and −31 bp relative to the TSS. The value of the Slide is about 0.4 Å. A shallow global minimum can also be seen between the positions −24 bp and −31 bp relative to the TSS. The Stiffness of the structure to the Roll alteration and to the Slide alteration in the non-coding promoters of both organisms are nearly constant with the exception of the INR element (Figure 4c,e). However, it should be noted that the Stiffness value to the Slide alteration is noticeably higher than the Stiffness value to Roll alteration. This is why the opening of the angles between adjacent base pairs towards minor groove leads to insignificant energy costs during TBP binding.

The profiles of the “Mobility to bend towards major groove” parameter of the non-coding promoters of both organisms differ most significantly from the random sequences profile (Figure 4f and Appendix A). Its maximum falls on the position −27 bp relative to the TSS. Thus, the values of the physical and structural parameters of non-coding promoters are, in general, similar to their values in the mRNA promoters, and the TATA-region stands out due to their mobility to bend towards major groove.

### 2.3. Local Variations of Ultrasonic Cleavage and DNase I Cleavage Intensities in Promoter Sequences

The sequence specificity of ultrasonic cleavage of DNA [24,25,26,27,28] as well as the sequence specificity of DNase I cleavage [29,30,31] allow us to study the genomic structures in more details.

We have discussed the coupling between the local intramolecular conformational movements in double-stranded DNA and intensities of ultrasonic cleavage C3′−O3′ bond in [10,13]. Indices of the intensities of ultrasonic cleavage of 16 dinucleotides (P) and 256 tetranucleotides (T) were previously obtained as a result of a detailed statistical studies of ultrasonic cleavage products with polyacrilamide gel electrophoresis [13,24,25,26,27,28]. We have used these indices earlier as well as their combination, S = (T − R)/R, to analyze the POL II core promoter regions of mRNA [10,11]. Index S gives information on the effects of the nearest context of dinucleotide on the intensity of its ultrasonic cleavage. If S < 0, the first and the fourth nucleotides of a tetranucleotide bring down the intensity of the cleavage in the central step; otherwise, they increase it.

The cutting rates of bovine pancreatic deoxyribonuclease I (DNase I) vary along a given DNA sequence, indicating that the enzyme recognizes sequence-dependent structural changes of the DNA double helix. The high-resolution crystal structures of two DNase I-DNA complexes showed that the enzyme binds tightly in the minor groove and to the sugar-phosphate backbones of both strands, thereby inducing a widening of the minor groove and the bending towards the major groove [29,30]. The context near the dinucleotide step strongly affects its cleavage efficiency. These can be rationalized by the fact that six base pairs are in contact with the enzyme. The intrinsic rate of cleavage by DNase I closely tracks the width of the minor groove [31]. We have used the indices of DNase I cleavage intensity at the hexanucleotide level (D), which were obtained in [32].

The profiles of the variation of ultrasonic cleavage indices and DNase I cleavage indices in the non-coding core promoters of *H. sapiens* and *M. musculus* are shown in Figure 5a–h and Appendix A, respectively. In the profiles, ultrasonic indices R, T, and S and the DNase I cleavage index D for the upper strand are depicted in blue, and in red for the lower (template) strand. All differences in the profiles are also shown in blue.

One can see that the slight attenuation of the ultrasonic cleavage (indices R and T) for both sets of non-coding core promoters is detected in the region from −31 to −24 bp relative to the TSS (Figure 5a,c and Appendix A). This indicates a slight decrease of conformational motion in this region. The profiles of the DNase I cleavage (index D) for both non-coding core promoters (Figure 5g and Appendix A) have local maximum at the TATA region for both strands. This indicates a minor groove widening. The S-indices differences profiles (Figure 5f and Appendix A) revealed periodic alternation of the intensities of conformational motion between complementary strands until the position of −5 bp. All profiles lose their smoothness and show a messy pattern of cleavage around the TSS.

Thus, the profiles of the local variations of ultrasonic cleavage as well as the profiles of the local variations of the structural and physical characteristics of naked DNA in core promoters of non-coding RNAs clearly show that the TATA box and INR regions differ significantly from the rest regions of core promoters.

## 3. Discussion

Earlier, we analyzed textual, physical, and structural characteristics of the mRNA core promoters of the fifteen organisms that belong to different steps of the evolutionary ladder. There were ten representatives of the animal kingdom—mammals, vertebrates and invertebrates, two representatives of the plant kingdom (*A. thaliana*, *Z. mays*), two representatives of the kingdom of unicellular fungi (*S. cerevisiae, S. pombe*), and a representative of Protozoa (*P. falciparum*). A special structural organization of double-stranded DNA in their core promoter regions was found, which contains two regions with singular mechanical and structural properties, namely TATA box and INR [10,11].

As some ncRNAs are transcribed by POL II, it was interesting to analyze the textual and structural characteristics of their promoters versus mRNA promoters. We have used datasets of *H. sapiens* and *M. musculus* promoters of lincRNAs and asRNAs. Our analysis has shown that the 3D structure of the naked DNA in the core regions of non-coding promoters is, in all respects, similar to the 3D DNA structure of mRNA core promoters. It also contains two regions with singular mechanical and structural properties, namely TATA box and INR.

The difference between promoters of mRNAs and non-coding promoters lies in their text characteristics. Informational entropy in each position of non-coding promoters in both organisms significantly exceeds the informational entropy of mRNA core promoters. This can lead to a noticeable decrease in the severity of “motives”, which are the characteristics of the nucleotide sequences of mRNA promoters that are important for the regulation of transcription [33].

There is significantly lower percentage of sequences in TATA box position of non-coding promoters that satisfy the TATAWAAR consensus than in TATA box position of mRNA promoters. Earlier crystallographic studies have shown [12] that TBP–DNA complex formation is permitted on many A/T-rich promoter sequences, albeit with dramatically reduced transcriptional efficiency in some cases. This may indicate that the kinetics of TBP binding to the TATA box, which influences transcription efficiency, is not the predominant criterion for the selection of nucleotide sequences for the core promoters of ncRNA. Their selection is based solely on their mechanical properties—those octanucleotides must expend as little intramolecular energy as possible when bending the double helix towards the wide groove.

Sequences in the TATA boxes of mRNA core promoters are partially selected in a similar way; however, their selection criteria, apparently, also contains the requirements for progeny reproducibility. It is well known that disease-associated single nucleotide substitutions (SNPs) influence the binding of target transcription factors. Exploration of the kinetic characteristics of the formation of human TBP complexes with TATA boxes; in which the SNPs are associated with β–thalassemia of diverse severity, immunosuppression, and neurological disorders; have shown, that hereditary diseases are largely caused by changes in TBP/TATA association rates [34].

## 4. Materials and Methods

**Database of nucleotide sequences:** we have analyzed the pooled sets of lincRNAs and asRNAs of POL II promoters that were retrieved from the first release of collections from the new section of the Eukaryotic Promoter Database (EPD) (http://epd.vital-it.ch (accessed on 24 October 2022)) [9]. We have used the sets of 2339 promoters of *H. sapiens* and 3077 promoters of *M. musculus*. The profiles of the averaged textual, structural, mechanical, and physicochemical properties of 80 bp core promoter sequences (positions from −50 to +30) were constructed. We have checked that all these sequences are 80 nucleotides long and strictly defined.

**Database of dinucleotide properties**: for analysis of the structural, mechanical, and physicochemical properties of the core promoter sequences, we have used indices of numerical parameterization for the ten double-stranded duplexes, which were collected from the DiProDB database (http://diprodb.fli-leibniz.de (accessed on 24 October 2022)) [17].

**Parameterization:** for the profile construction of the stacking energy and the base-pair step parameters Roll and Slide, we have used the parameterization of Perez et al. [21]; for the profile construction of the stiffness variation of DNA double helix to the Roll and Slide changes, we have used the parameterization of Goni et al. [22]; and for the profile construction of stiffness of the structure to bend towards major groove, we have used the parameterization of Gartenberg and Crothers [23].

**Profiles construction:** X-axes of the profiles define the position relative to the TSS. This was denoted as +1 bp, while negative and positive numbers denote upstream and downstream regions. Y-axes present the mean value of a chosen characteristic from the corresponding databases. For textual characteristics, defined at the mononucleotide level, for every 80 positions on the X-axis (numbered: −50,−49, …. −1,+1,+2,…+30), the amounts of each type of nucleotides (A, C, G, T) in all core promoters from a set of chosen species are summed up, and the resulting sum is divided by the number of promoters. For physical or structural characteristics, defined at the base-pair step level, or for ultrasound cleavage rates at the dinucleotide level, for every 79 positions on the X-axis (numbered: −49,−48,….−1,+1,+2, …+30), the values of these characteristics are summed up (for dinucleotides at the positions [(−50,−49); (−49,−48);…(−1,+1); …(+29,+30)], which are taken from DiPro DB or Table 1 in [14], and the resulting sum is divided by the number of promoters. For ultrasound cleavage rates at the tetranucleotide level, for every 77 positions on X-axis (numbered: −48,−47,….−1,+1,+2,…+29), the values of these characteristics for tetranucleotides are summed up (for tetranucleotides at the positions (−50,−49,−48,−47); (−49,−48,−47,−46); …(−2,−1,+1,+2); …(+27,+28,+29,+30)), which are taken from Table S1 in [14], and the resulting sum is divided by the number of promoters. For DNAse I cleavage rates at the hexanucleotide level, for every 75 positions on the X-axis (numbered: −47,−46, ….−1,+1,+2,…+77,+78), the values of these characteristics are summed up (for hexanucleotides at the positions (−50,−49,−48,−47,−46,−45); (−49,−48,−47,−46,−45,−44); …(−3,−2,−1,+1,+2,+3); …(+25,+26,+27,+28,+29,+30), which are taken from Supplementary to the work [32], and the resulting sum is divided by the number of promoters.

We have written the programs in Python 3.10 for profiles construction.

## 5. Conclusions

The organization of the 3D structure of the naked DNA in POL II core promoter regions of lincRNAs and asRNAs is, in many respects, similar to the organization of the naked DNA in mRNA core promoters. Namely, their 3D structure contains two regions with singular mechanical and structural properties: TATA box and INR.

Nevertheless, a number of differences were identified in the text characteristics between the core promoters of pooled sets of lincRNAs and asRNAs as for *H. sapiens* as for *M. musculus*, and the sets of mRNAs core promoters of these organisms. These defferences are following: (a) the informational entropy in each position of the nucleotide sequences of pooled samples of lincRNAs and asRNAs significantly exceeds the informational entropy of mRNA core promoter sequences; (b) among octanucleotides that represent the TATA box in the sets of core promoters of ncRNAs, the percentage of sequences that satisfy the TATAWAAR consensus is markedly lower than in the sets mRNA core promoters.

By our opinion these textual differences in the core promoter regions may lead to the spontaneity and lower level of their transcription compared to mRNA transcription.

## Figures and Tables

**Figure 1 ijms-24-08199-f001:**
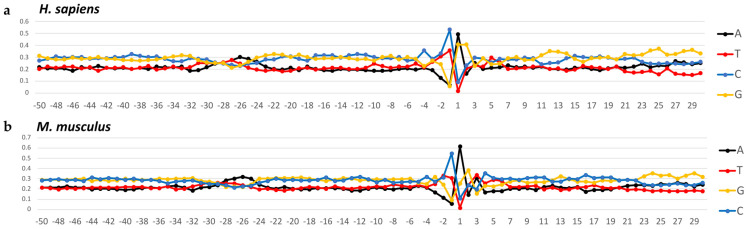
Profiles of core promoter sequences as the mononucleotide frequencies of occurrence (in percentages) at each position along the strand, complementary to template (**a**) for data sets of *H. sapiens* and (**b**) for data sets of *M. musculus*.

**Figure 2 ijms-24-08199-f002:**
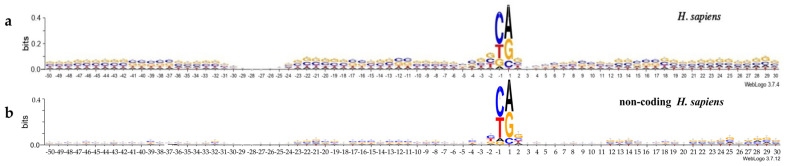
Logo representation with information content of 0.4 bit of the (**a**) mRNA promoter sequences of *H. sapiens* and (**b**) non-coding promoter sequences of *H. sapiens*.

**Figure 3 ijms-24-08199-f003:**
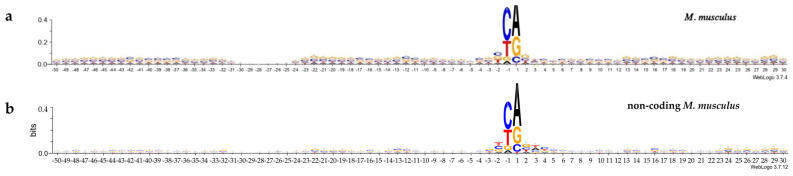
Logo representation with information content of 0.4 bit of the (**a**) mRNA promoter sequences of *M. musculus* and (**b**) non-coding promoter sequences of *M. musculus*.

**Figure 4 ijms-24-08199-f004:**
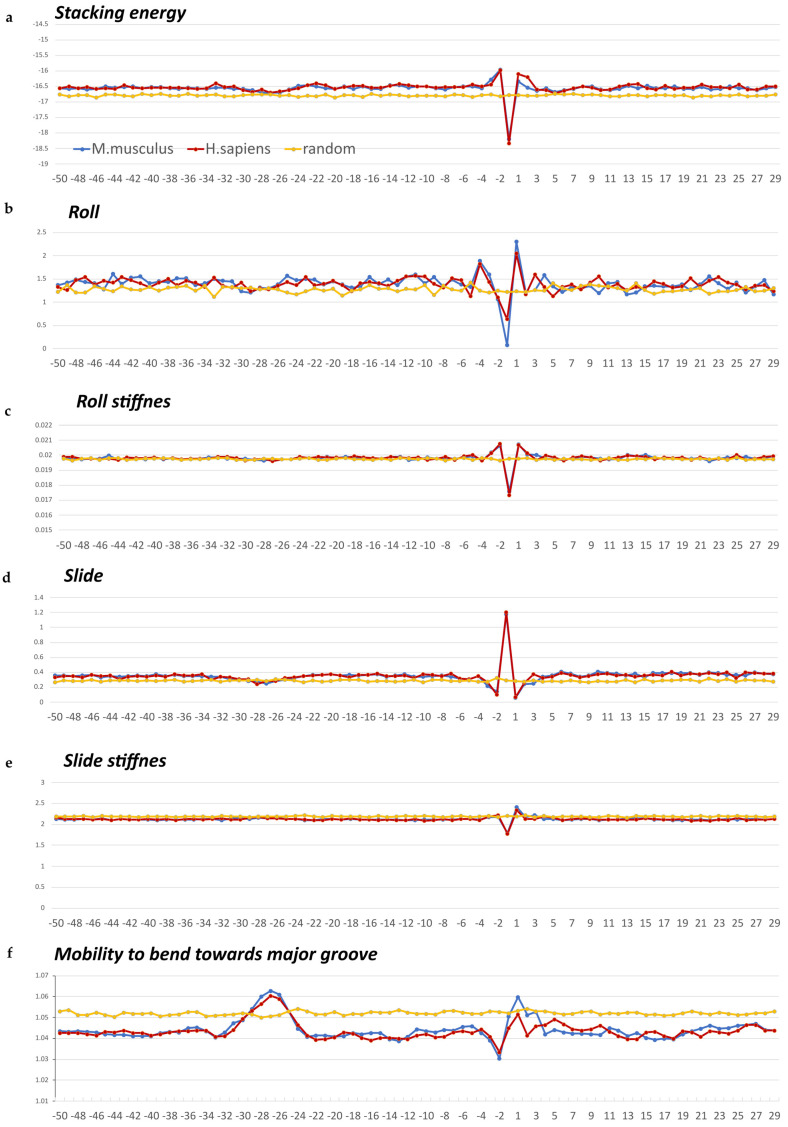
(**a**–**f**) Local variations of the values of physical and structural parameters in core promoter regions of non-coding promoters of *H. sapiens* (in red) and *M. musculus* (in blue) along with the profile of the 80 bp set of 3000 computer-simulated random nucleotide sequences (in yellow). (**a**) Stacking energy (in kcal/mol). (**b**) Roll (in degrees). (**c**) Stiffness of the duplex structure to roll alteration (in kcal/mol degree). (**d**) Slide (in angstroms). (**e**) Stiffness of the duplex structure to slide alteration (in kcal/mol angstrom). (**f**) Mobility to bend towards major groove (in mobility units).

**Figure 5 ijms-24-08199-f005:**
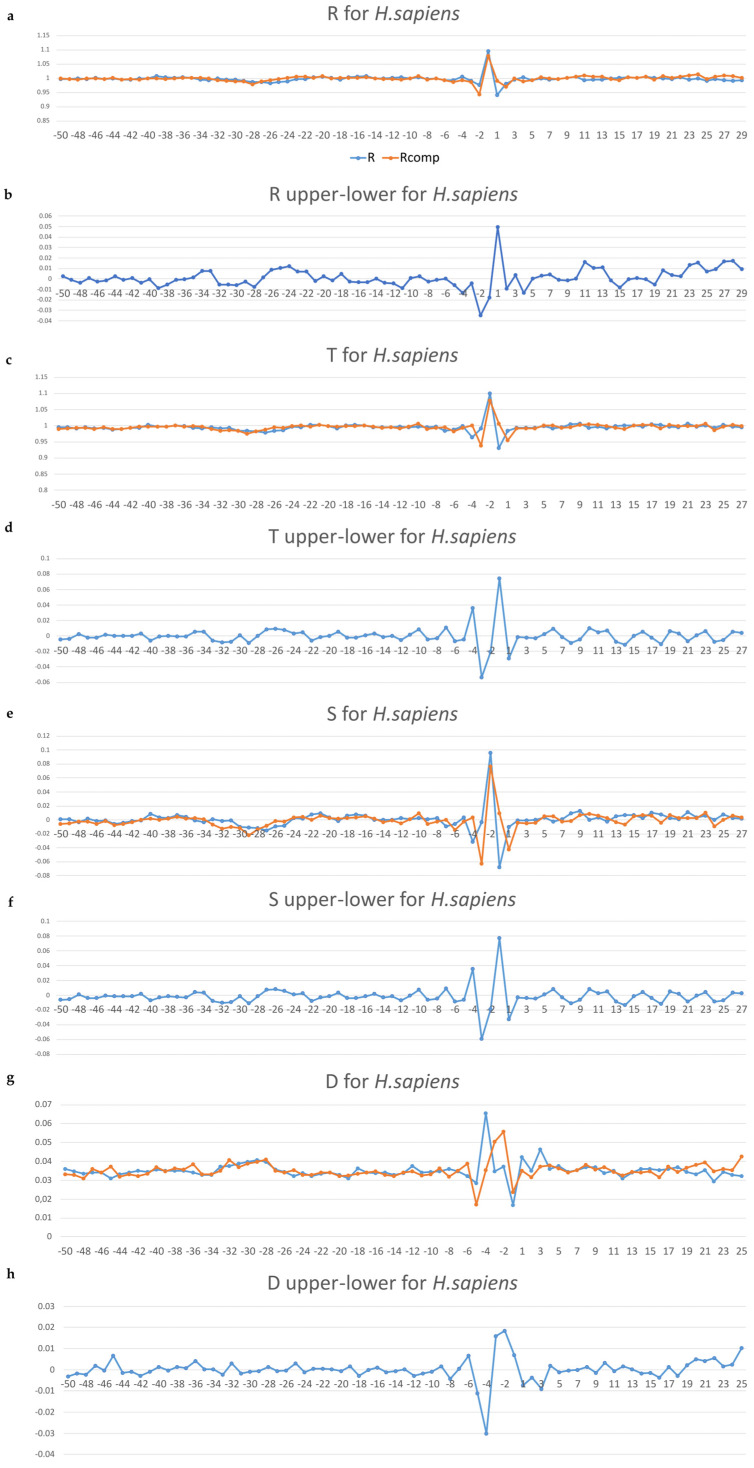
(**a**–**h**) (**a**) Profiles of the relative intensities of ultrasonic cleavage of 16 dinucleotides (P); (**b**) profiles of their differences between complementary strands (P_upper_ − P_lower_); (**c**) profiles of the relative intensities of ultrasonic cleavage of 256 tetranucleotides (T); (**d**) profiles of their differences between complementary strands (T_upper_ − T_lower_); (**e**) profiles of indices S = (T − R)/R, (**f**) profiles of their differences between complementary strands (S_upper_ − S_lower_); (**g**) DNase I cleavage indices at hexanucleotide level of resolution (D); (**h**) profiles of their differences between complementary strands (D_upper_ − D_lower_) for *H. sapiens* non-coding promoters.

**Table 1 ijms-24-08199-t001:** Frequencies of occurrence of different octanucleotides in the TATA box of non-coding promoters of *M. musculus* and *H. sapiens*.

	*M. musculus* (−31: −24)		*H. sapiens* (−31: −24)	
1	TTTTTTTT	0.42%	GTTTATCA	0.47%
2	ACAATATA	0.16%	AGAATAAA	0.38%
3	GTATAAAA	0.13%	TTTTATAA	0.26%
4	TAGTTATG	0.13%	TTTGTTTA	0.21%
5	TTTAAAAG	0.13%	CTATAAAG	0.17%
6	AATAAAAG	0.13%	CCCGGAAG	0.17%
7	CTATAAAA	0.13%	CCCCGCCC	0.17%
8	AGGGTAAA	0.10%	TATTTAAA	0.17%
9	AACAGGAG	0.10%	TATAAAAG	0.17%
10	AATAGCAG	0.10%	CTATAAAA	0.17%
11	CCTTGCTA	0.10%	GCCTTGCA	0.13%
12	GTATATAA	0.10%	TATATAAA	0.13%
13	CCCGCCCG	0.10%	GAGCTAAT	0.13%
14	TTAAGACC	0.10%	CCTTAAAA	0.13%
15	AAATAAAT	0.10%	CCCCTCCC	0.13%
16	AGAGAGAG	0.10%	GTTGAGGT	0.13%
17	TATAAATA	0.10%	CCATGCAG	0.13%
18	TCTATAAA	0.10%	GGGCGGGG	0.13%
19	AGCGCGCG	0.10%	TATTTATA	0.13%
20	TATATAAG	0.10%	TTAAATAG	0.09%

**Table 2 ijms-24-08199-t002:** Frequencies of occurrence of octanucleotides in the sets obtained while imposing the condition “*restricting the selection to promoters that contain a TATA box*”.

	*M. musculus* (−31: −24)		*H. sapiens* (−31: −24)	
1	ACAATATA	1.61%	AGAATAAA	3.60%
2	AATAAAAG	1.29%	TTTTATAA	2.40%
3	TTTAAAAG	1.29%	CTATAAAA	1.60%
4	GTATAAAA	1.29%	TATAAAAG	1.60%
5	CTATAAAA	1.29%	TATTTAAA	1.60%
6	TATAAAAG	0.96%	CTATAAAG	1.60%
7	ATATAAGG	0.96%	TATATAAA	1.20%
8	AAATAAAT	0.96%	CCTTAAAA	1.20%
9	GTATATAA	0.96%	TATTTATA	1.20%
10	TATTTATT	0.96%	TTATAAAG	0.80%
11	AATAAAAA	0.96%	CCTTTAAA	0.80%
12	TATATAAG	0.96%	ATAAAAAC	0.80%
13	TTTAAAAC	0.96%	GGATAAAA	0.80%
14	TATAAATA	0.96%	TTTAAAGG	0.80%
15	CTATTTAG	0.96%	GTAGAAAA	0.80%
16	TATAAAAA	0.64%	CCCATAAA	0.80%
17	TATAAATG	0.64%	TTTATAAA	0.80%
18	CTATATAA	0.64%	GAAATAAA	0.80%
19	ATAAAAGA	0.64%	TTTTAAAA	0.80%
20	TTTAAAAA	0.64%	CTATAAAT	0.80%

**Table 3 ijms-24-08199-t003:** Frequencies of occurrence of dinucleotides PyPu, PuPu, PyPy, and PuPy in the positions (−1, +1) of INR.

	PyPu	PuPu	PyPy	PuPy
*H. sapiens*	79.86%	9.66%	8.59%	1.84%
*M. musculus*	76.02%	10.98%	9.23%	3.74%

**Table 4 ijms-24-08199-t004:** Frequencies of occurrence of dinucleotides in the positions (−1, +1) of INR.

	*M. musculus*	*H. sapiens*
CA	42.77%	34.67%
TG	13.94%	20.52%
TA	12.58%	11.54%
CG	6.73%	13.13%
CC	4.29%	4.53%
GA	4.22%	2.27%
TC	4.06%	3.04%
GG	2.66%	1.75%
AA	2.11%	0.68%
AG	1.98%	4.96%
GC	1.49%	0.94%
AC	1.27%	0.51%
CT	0.71%	0.77%
GT	0.55%	0.30%
AT	0.42%	0.09%
TT	0.16%	0.26%

## Data Availability

Not applicable.

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
