# Peer review of "Core Promoter Regions of Antisense and Long Intergenic Non-Coding RNAs"

_ijms, 2023, doi:10.3390/ijms24098199_

Round 1

Reviewer 1 Report

Savina et al. realized a very interesting article describing the “Core promoter regions of antisense and long intergenic non-coding RNAs”. I consider the manuscript very interesting but, at the same time, I suggest several revisions needed to improve the reliability and the completeness of the paper:
•    The “Introduction” section should be more updated and improved. I suggest adding data related to post-transcriptional and post-translational modifications. The recent PMID: 32184807 and PMID: 36290689 could represent a substrate able to enforce the role of considered cellular mechanisms.
•    The “Results” section is too long and, above all, presents too figures and tables. I suggest to shift some of them to Supplementary Materials, referring to them throughout the text.
•    The “Discussion” section should be cleared in the final part.
•    The “Methods” section should be improved in the first part, probably dividing it into paragraphs, like the following “Profile construction”.
•    Finally, manuscript requires important English revisions and typos correction.

Reviewer 2 Report

The authors explored the promoter sequences for lncRNAs and found textual differences between lncRNA and mRNA. The comparison and results are interesting, though I find the presentation of the results difficult to make sense of. Also, I disagree with the way inter-species comparisons are made; if one were to compare the promoter regions between species, it should be limited to only the lncRNA that are present in both species.

Minor issues:

- Line 107: The sentence is fragmented. Do you mean "the result is in line with what we obtained from ..."

- Line 114-115:Unsure what is meant by "As it is seen from the Table 1 the consensus sequences in TATA position of non-coding promoters we meet extremely rare"

- Line 115-116: Unsure what the sentence meant.

- Line 122-123: Please rephrase. Meaning unclear.

- Line 124: "We have get" is incorrect.

- Line 129: "This sample" has been referred to a few times. I am unsure what it means - is this result from one sample? I thought a few thousands of samples were included (line 64-65)

- Line 160-161: Please rephrase. I think you meant "preferred". 

- Line 162: "The occurrence of octanucleotides," - please remove the comma.

- Line 169: "ofeach" missing space between the words.

- Line 186: Please state what PyPu, PuPu...etc are.

- Tables (all): I don't think you need to say "in percentage" or "in percent's" since the percentage marks is in the table.

- Figures (all): the font size is too small for reading.

- Figures (all): It would be good to see side by side comparison between mRNA and non-coding RNA

- Figure 4: Why is there a vertical line in some of the plots?

- Should be "homo sapiens" rater than "H. sapience"

- Grammatical issues - please re-read and edit.

- The authors compared inter-species differences for the promoter region, but that is based on all lncRNA; this should be done on only the lncRNA that share homology.
